# Machine learning-based phenotypic imaging to characterise the targetable biology of *Plasmodium falciparum* male gametocytes for the development of transmission-blocking antimalarials

Oleksiy Tsebriy[1☉], Andrii Khomiak[1☉], Celia Miguel-Blanco[2], Penny C. Sparkes[3], Maurizio Gioli[4☉], Marco Santelli[1], Edgar Whitley[5☉], Francisco-Javier Gamo[2], Michael J. Delves[3]*

1 Independent researcher, Ternopil, Ukraine, 2 Global Health Medicines R&D., GlaxoSmithKline, Madrid, Spain, 3 Department of Infection Biology, London School of Hygiene and Tropical Medicine, Keppel Street, London, United Kingdom, 4 SOAS University of London, London, United Kingdom, 5 Department of Management, London School of Economics and Political Science, London, United Kingdom

☉ These authors contributed equally to this work.
* michael.delves@lshtm.ac.uk

**Data Availability Statement:** All imaging datasets used in this study are accessible at: https://www.

## Abstract

Preventing parasite transmission from humans to mosquitoes is recognised to be critical for achieving elimination and eradication of malaria. Consequently developing new antimalarial drugs with transmission-blocking properties is a priority. Large screening campaigns have identified many new transmission-blocking molecules, however little is known about how they target the mosquito-transmissible *Plasmodium falciparum* stage V gametocytes, or how they affect their underlying cell biology. To respond to this knowledge gap, we have developed a machine learning image analysis pipeline to characterise and compare the cellular phenotypes generated by transmission-blocking molecules during male gametogenesis. Using this approach, we studied 40 molecules, categorising their activity based upon timing of action and visual effects on the organisation of tubulin and DNA within the cell. Our data both proposes new modes of action and corroborates existing modes of action of identified transmission-blocking molecules. Furthermore, the characterised molecules provide a new armoury of tool compounds to probe gametocyte cell biology and the generated imaging dataset provides a new reference for researchers to correlate molecular target or gene deletion to specific cellular phenotype. Our analysis pipeline is not optimised for a specific organism and could be applied to any fluorescence microscopy dataset containing cells delineated by bounding boxes, and so is potentially extendible to any disease model.

ebi.ac.uk/biostudies/bioimages/studies/S-BIAD633/ All software code used in this study is available at: https://github.com/michaeldelves/PhIDDLI/

**Funding:** This work was supported by a Bloomsbury SET award (BSA28) to MJD, EW and MG; a Wellcome Institutional Translational Partnership Award (214227/Z/18/Z) to MJD; and a Medical Research Council Career Development Award (MR/V010034/1) to MJD. PCS is supported by an MRC-LID studentship. The funders had no role in study design, data collection and analysis, decision to publish, or preparation of the manuscript.

**Competing interests:** The authors have declared that no competing interests exist.

## Author summary

Interventions that prevent malaria parasite *Plasmodium falciparum* from transmitting from humans to mosquitoes are highly desirable to prevent both the spread of malaria and crucially the spread of drug resistance. Transmission is caused by the non-pathogenic gametocyte stage of the parasite life cycle which are insensitive to most current antimalarials. Consequently new drugs and new drug targets need to be identified to meet this need. Here we present PhIDDLI–a machine learning image analysis pipeline which we use to characterise the phenotype of male gametocytes treated with 40 novel transmission-blocking molecules. We found that the molecules formed 5 distinct phenotypic clusters according to their putative timing/mode of action within the cell. Molecules with similar chemical structures gave similar phenotypes. Interestingly, by varying the timing of drug exposure, different phenotypes manifest between molecules with seemingly similar initial phenotypes. Our study provides the first insights into the breadth of drug-targetable cell biology in male gametocytes and provides reference images and tool compounds for the identification and validation of new transmission-blocking drug targets.

## Introduction

*Plasmodium*, the causative agent of malaria, has a complex life cycle involving both a vertebrate host and mosquitoes. Whilst the asexual stages of the parasite life cycle cause disease pathology, transmission to the mosquito is exclusively mediated by the non-pathogenic gametocyte stage. In *Plasmodium falciparum*, the most deadly human-infective species, gametocytes develop to maturity (mosquito infectivity) over a period of 8–10 days [1]. During this five-stage developmental process, gametocytes lose sensitivity to most approved antimalarial drugs [2]. This allows mature gametocytes in malarial patients (and asymptomatic carriers) to escape standard drug treatment, continue transmission, and propagate any drug resistance mutations that have potentially developed within the human host.

Drug treatments that prevent transmission by either targeting the gametocyte directly or preventing their onward infectivity to the mosquito are a priority [3], with primaquine, tafenoquine, methylene blue, and atovaquone being the only approved drugs with potent transmission-blocking (TB) properties–each with their own difficulties that limit their widespread implementation.

Over the past decade, there have been many screening campaigns to identify TB molecules using a variety of *in vitro* assays [2,4–7]. Whilst this work continues, there are still significant gaps in our knowledge as to how TB molecules act, and which proteins and pathways they target.

Impressive success has been made with identifying the targets and mechanisms of resistance of new antimalarial molecules targeting the asexual stages of the parasite using drug resistance generation and whole genome sequencing [8]. Unfortunately, given the non-replicative nature of gametocytes, this methodology is not viable to characterise transmission-blocking molecules. Recently, photo-affinity labelling has been successfully used to propose a protein target of one class of transmission-blocking molecules—the N-((4-hydroxychroman-4-yl)methyl)-sulfonamides (N-4HCS) that target Pfs16 [9]. However, this approach is laborious, expensive, and limited by the individual chemistry of the molecule under investigation.

In a previous high-throughput screen of compounds targeting the functional viability of male and female *P. falciparum* gametocytes, it was observed that different molecules targeting male gametocytes manifest distinct phenotypes when visualised by antibody labelling and

fluorescence microscopy [6]. Whilst not definitively identifying a mode of action *per se*, understanding the action and characterising the activity of transmission-blocking molecules is a critical first step in uncovering the underlying biology targetable in gametocytes [10]. This will also help to prioritise which molecules should progress forward in drug development, and identify potential partner drugs for transmission-blocking combination therapies that target different pathways to mitigate drug resistance.

Fluorescence microscopy and automated image analysis using artificial intelligence (AI) and machine learning (ML) techniques have already been widely adopted for drug screening and both genetic and RNA interference (RNAi) screens [11]. The use of AI/ML enables researchers to analyse many cells rapidly and in an unbiased manner to assign phenotypes, categorise, and compare different treatment conditions. For malaria, many different AI/ML approaches have been implemented to identify parasite-infected erythrocytes from thin blood smears for diagnostics [12–14] and, recently, AI/ML has been used to investigate the phenotypic effects of several antimalarial drugs on asexual parasites [15].

Here we have developed a simple, yet powerful AI/ML pipeline called Phenotype Image Data and Digital Learning Innovation (PhIDDLI) comprised of "off the shelf" software modules with permissive software licences permitting modification and free distribution. Using PhIDDLI we characterise the activity of a panel of 40 transmission-blocking molecules, many of which possess unknown modes of action and make available all code and images to enable further study and modifications.

## Results

### 1. Identification of molecules targeting the functional viability of male and female *P. falciparum* gametocytes

A subset of 564 molecules from the Tres Cantos Antimalarial Set (TCAMS) with established activity against *P. falciparum* asexual stage parasites [16] was screened at 2μM for additional transmission-blocking activity in the *P. falciparum* Dual Gamete Formation Assay (PfDGFA) [17]. In total, 262 molecules were found to give >50% inhibition against either male gametocytes, female gametocytes, or both, in at least three of four biological replicates (**S1 Table**). Of these, 197 had $IC_{50}$ values ≤10μM in either the male or female assay readout (or both) (**S2 Table**). By counter-screening against HepG2 cells to investigate general cytotoxic activity, 80 molecules were determined to have greater than ten-fold cytotoxicity index therefore deemed parasite-selective, and thus were progressed (**S3 Table**). Previously published HepG2 cytotoxicity data [5,7,18] was integrated into the analysis and by taking the mean $TOX_{50}$ values from all available data, an additional seven previously discounted molecules were "rescued" and two molecules discounted from the PfDGFA screen, giving 85 molecules selected for further study.

Although many of these molecules have been identified in other transmission-blocking screens, 26 were unique to this study (**Fig 1A**). In line with previously observed trends [19,20], very few molecules were observed to be more active against female than male gametocytes, with the most female-selective molecule TCMDC-125849 being 4.7 times more potent against females than males (**S3 Table**). In contrast, 52 molecules were substantially more active against male than female gametocytes (**Fig 1B**).

To understand the chemical diversity of the identified transmission-blocking molecules, they were clustered by FragFp similarity (**Fig 2**). A diverse range of chemical scaffolds was found to be present, with some scaffolds represented by several molecules. For example, five compounds (TC04, TC13, TC27, TC41 and TCMDC-140369) possessed a trisubstituted imidazole scaffold which has recently been reported to inhibit the activity of *P. falciparum* protein

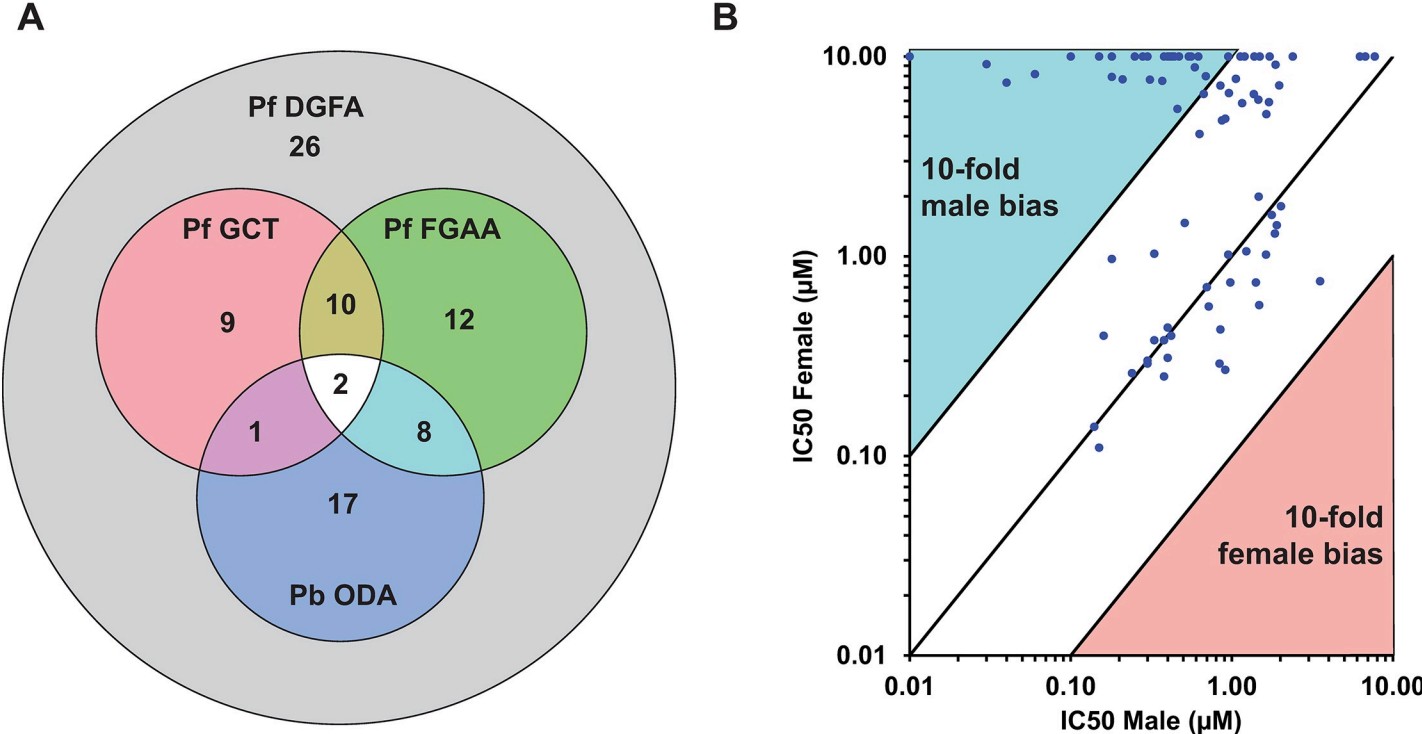

**Fig 1. The activity of TCAMS compound identified in the Pf DGFA screen.** (A) 85 molecules were confirmed active in the Pf DGFA. 58 were previously identified in one or more transmission-blocking screens. (B) The majority of molecules displayed similar activity against male and female gametocytes, or biased/specific activity against male gametocytes. No molecule was identified with >4.7-fold greater activity against female gametocytes. Abbreviations (IC50 = concentration giving 50% inhibition, PbODA = *P. berghei* Ookinete Development Assay [5], PfDGFA = *P. falciparum* Dual Gamete Formation Assay, PfFGAA = *P. falciparum* Female Gametocyte Activation Assay [7], PfGCT = *P. falciparum* Gametocyte Viability ATP Assay [18]).

kinase G (PfPKG) [21]. Indeed, TCMDC-140369 identified here is structurally identical to previously reported MMV030084 [21]. A literature search was performed to identify putative targets for the molecules based upon existing characterisation and the established activity of close analogues [16,22]. Whilst it was not possible to discover evidence for a target for most of the molecules, 21 were identified as putative kinase inhibitors, two may target protein synthesis, and one is proposed to be a putative inhibitor of microtubule organisation (**Fig 2** and **S3 Table**).

## 2. Phenotypic activity against male gametogenesis

Male gametogenesis is a complex process that involves cell shape change (i.e., "rounding up" from a falciform shape), three rounds of endomitotic division, formation of ordered microtubule-rich flagella, egress from the erythrocyte, and finally emergence of up to eight flagellated male gametes [23]. Many transmission-blocking molecules do not kill gametocytes directly, rather they render the gametocytes infertile (i.e., sterilise) or interfere directly with the process of gametogenesis (i.e., contraceptive) [19]. Previous studies have shown that male gametocytes treated with different transmission-blocking molecules show markedly distinct cellular phenotypes when triggered to undergo gametogenesis *in vitro* [6]. This indicates that there are multiple different biological processes essential to male gametocytes that are targetable by small molecules. However, the breadth of this drug-targetable cell biology is unknown. To shed light on this and aid the development of evidence-based hypotheses for the mode of action of

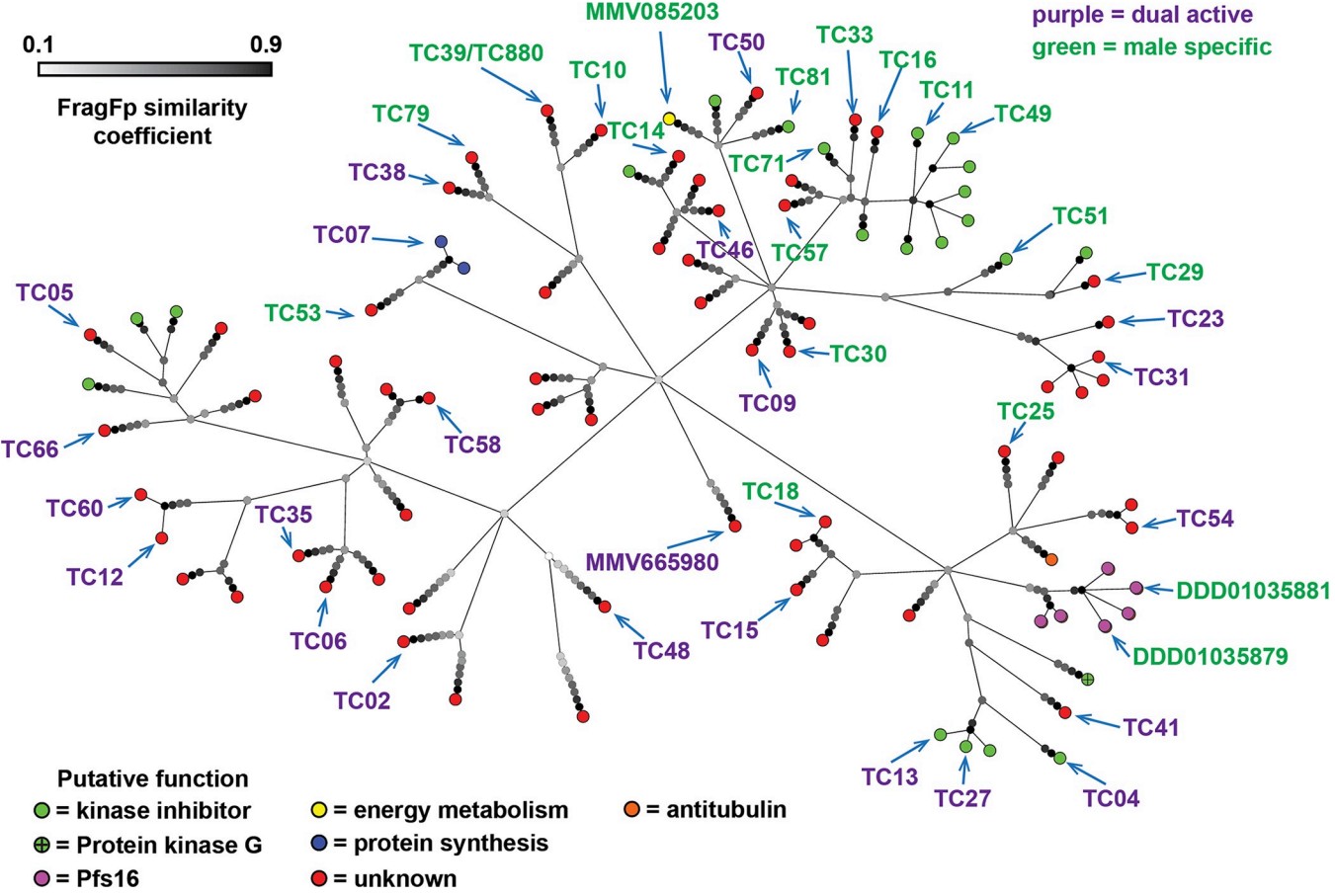

**Fig 2. The molecules active against *P. falciparum* gametocytes identified in this study clustered based upon chemical structure similarity using the FragFp algorithm.** Terminating nodes at the ends of branches represent compounds and their colour indicates putative function based upon literature and database annotation of similar compounds. The greyscale colour of nodes at branch points indicates similarity of compounds. Dark branchpoints = highly similar; Light branchpoints = divergent chemical structures. Compounds labelled with codes were studied in more detail. The colour of compound name indicates whether it was found to specifically target male gametocytes or target both gametocyte sexes in the initial Pf DFGA screen.

transmission-blocking antimalarial compounds, 40 molecules were selected from the 85 Pf DGFA actives for phenotypic characterisation.

Candidate selection was based upon compound availability and molecules demonstrating at least a ten-fold selectivity index in the Pf DGFA compared to cytotoxic activity against HepG2 cells (S3 Table). The testing pool was further increased with a panel of existing established antimalarial compounds with previously reported evidence of their mode of action: DDD01035881 and DDD01035879 are N-((4-hydroxychroman-4-yl)methyl)-sulfonamides (N-4HCS) that appear to affect male gametocyte microtubule dynamics [10] and have recently been proposed to be inhibitors of Pfs16 [9]. MMV665980 that has been found to inhibit coenzyme A biosynthesis [24]. MMV085203 has recently been reported to target energy metabolism in the parasite mitochondrion [25]. ML10 is a potent inhibitor of PfPKG—a kinase that is essential for the initial signalling cascade that commences gametogenesis [26]. Finally, Methylene Blue is an approved antimalarial which is thought to affect the redox balance within the parasite [27].

Mature gametocytes were treated for 24h with approximately $10xIC_{50}$ of test compounds or DMSO as a solvent control. Male gametogenesis was then induced by lowering the

temperature from 37˚C to ~20˚C and adding xanthurenic acid. 20min later, when male gametogenesis (exflagellation) could be expected to peak, all samples were fixed. A sample of noninduced DMSO-treated gametocytes was fixed with 4% paraformaldehyde to provide an unactivated gametocyte control for comparison. The cells were then processed for immunofluorescence using an antibody against alpha-tubulin II and DNA staining with 4′,6-diamidino-2-phenylindole (DAPI) and visualised by fluorescence microscopy. Tubulin-bright cells from each treated sample were imaged and captured at random.

## 3. Identification of cells and PhIDDLI classification

Cells were identified using a semi-automated pipeline established in the ICY BioImageAnalysis (https://icy.bioimageanalysis.org/) protocols module. Images and accompanying bounding boxes representing the outlines of identified cells were then passed to the PhIDDLI pipeline. PhIDDLI consists of several open source image processing and AI/ML modules connected for seamless processing, clustering and visualisation of cellular phenotypes (**Fig 3A**). Images (n = 1207) and bounding boxes were accepted by the YOLOv5 (https://ultralytics.com/yolov5) module for onward formatting and extraction of individual cells (n = 2446) from images. To focus on the overall cellular phenotype rather than potentially confounding variable immunostaining intensities between experimental replicates, cells were transformed so that the range of staining intensities for each channel were normalised to fall between a 0–255 pixel intensity value. Then, using the Albumentations module (https://albumentations.ai/) to augment the dataset and control for cells in different orientations, each extracted cell image was rotated 90, 180 and 270 degrees and mirrored so that each cell was represented eight different times in the dataset. The augmented images were then passed to the EffcientNet-B1 module (https://github.com/tensorflow/tpu/tree/master/models/official/efficientnet) (a pretrained neural network developed for image processing) which extracted 1,000 distinct numerical features from each individual cell image. The EfficientNet-B1 was selected from amongst the other EfficientNet models due to its input image size (240 pixels) best matching the size of the individual cell images in the dataset, therefore ensuring maximal input data whilst allowing acceptable computation time on a desktop computer. The dimensionality of these features was then reduced using *t*-distributed stochastic neighbour embedding (t-SNE) (https://scikit-learn.org/stable/modules/manifold.html#t-sne) and clustered using k-means clustering. The output was then automatically visualised as an interactive graph using a Plotly Dash module (https://plotly.com/dash/) which displays t-SNE distribution, cluster assignment and overlays the original identified cells (**Fig 3B**).

To understand if experimental variability affected the ability of PhIDDLI to process and classify individual cells over different replicates, the control cells from all replicates were compared. Across six replicates (**Fig 4A**), unactivated cells were all located in the upper region of the t-SNE distribution. Contrastingly, the DMSO-treated cells that were allowed to progress through gametogenesis were almost completely excluded from this region. These cells formed a tight focal point central in the t-SNE distribution. The interactive PhIDDLI visualisation tool confirmed that this region contained the exflagellating cells. In addition there were other DMSO-treated cells in the lower-right section of the distribution. Examination of these cells confirmed that these represented cells that had progressed late into male gametogenesis but arrested before exflagellation.

Nine phenotypic clusters were identified by the pipeline, and using the interactive visualisation tool it was possible to visualise the cells within each cluster (**Fig 4B**). There were several clusters with visibly similar appearance, however these neighboured each other and likely represent a continuum of phenotypes rather than distinct isolated groups. Clusters 1 and 3 clearly

**A**                                                              **B**

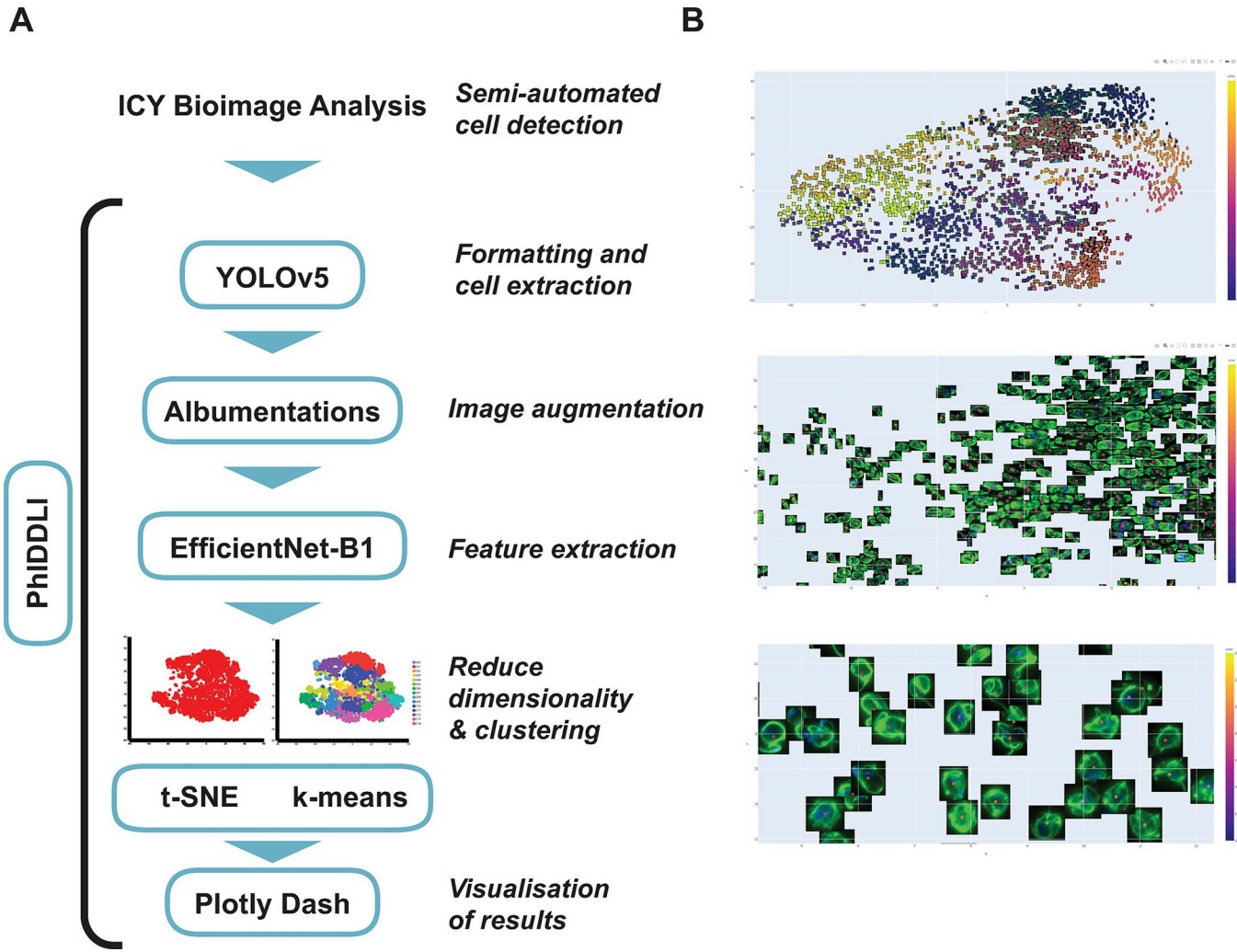

**Fig 3. Using phenotypic imaging to cluster drug-treated cells with similar phenotypes.** (A) Raw fluorescence microscopy images were processed in ICY Bioimage Analysis to identify cells and then passed to the PhIDDLI pipeline for machine learning based clustering. (B) PhIDDLI interactive clustering output showing the distribution of cell phenotypes and navigable to view individual cells within each cluster.

represented falciform gametocytes with diffuse microtubule staining and diffuse DNA label-ling–all indicative of gametocytes that failed to commence gametogenesis. Clusters 2 and 7 appeared to contain cells that had undergone the initial rounding up process of male gameto-genesis, but retained small concentrated nuclear staining suggestive of arrest prior to DNA replication, and relatively diffuse microtubule staining indicating lack of flagellar organisation. Cluster 0 contained cells that appeared to be shaped somewhere between falciform and round, with tubulin staining mostly around the periphery and diffuse DNA staining. Cluster 8 con-tained cells with defined, strong tubulin staining but diffuse DNA labelling suggesting forma-tion of some microtubule structures in the absence of DNA replication. Clusters 4 and 5 were similar, containing cells with defined microtubule structures and large, concentrated DNA staining. These cells appeared to have replicated their DNA and progressed with flagellum for-mation, but arrested short of gamete emergence from the cell. Cluster 6 was located at the con-fluence of most other clusters and hence possessed a range of cellular phenotypes that generally included cells with irregular shape (S1 Fig). The predominant phenotype in cluster 6

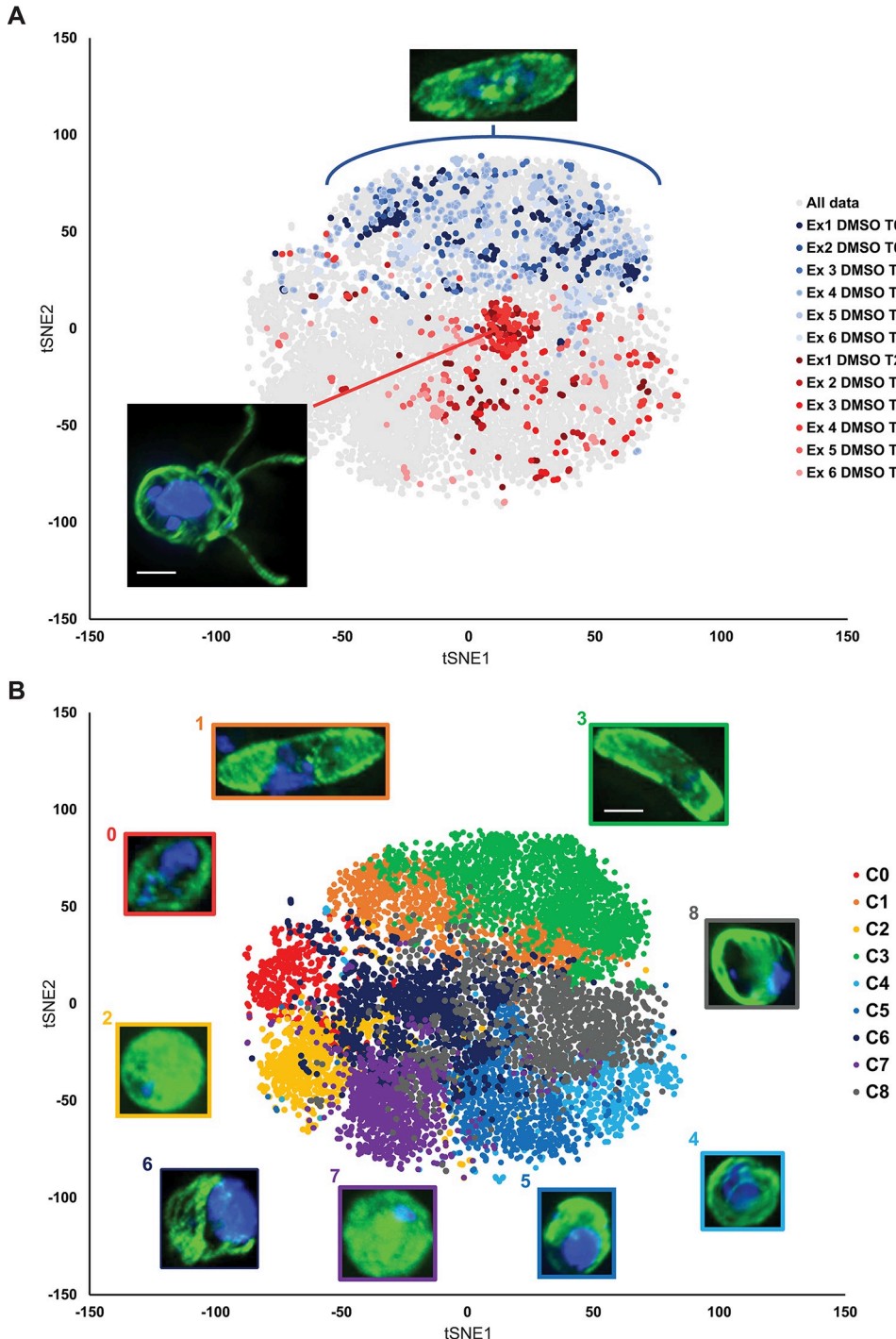

**Fig 4. Phenotypic distribution of male gametocytes treated with 45 transmission-blocking compounds.** 2446 cells (augmented in eight different orientations) from 45 different treatments/controls were analysed by PhIDDLI and visualised by t-SNE and k-means clustering into nine clusters. (A) Localisation of the unactivated control samples (blue shades) and untreated control samples (red shades) within the dataset. Different shades indicate control cells from the six independent experiments comprising the entire screen. (B) The analysed dataset coloured by computed cluster identity. Images show representative cells from each cluster. The colour of the border surrounding the cell matches the cluster it represents. Scale bars = 3μm.

was cells with clearly defined internal microtubule structures and large, bright areas of DNA staining, suggestive of male gametocytes immediately prior to emergence of male gametes. Supporting this assignment, within the cluster, as it approached the centre of the t-SNE distribution frequently included cells with one or more gametes emerging.

## 4. Phenotypic clustering of test molecules

As cells were imaged at random in the drug-treated samples, a particular treatment could be expected to possess cells with a range of cellular phenotypes due to the fact that *in vitro*, male gametogenesis is an inefficient process that not every male gametocyte completes after stimulation. Nevertheless, visual inspection of the distribution of cells treated with a particular molecule clearly demonstrated non-random distribution (S2 Fig). To understand the distribution of phenotypes within particular drug treatments, the percentage of cells in each treatment that fell into each phenotypic cluster was calculated and compared by principal component analysis (PCA) and k-means clustering (S3 Fig).

This clearly separated molecules by their phenotype, identifying five phenotypic clusters (**Fig 5**). Molecules that prevented total activation of male gametogenesis were all found in cluster A. Cluster D possessed molecules that appeared to arrest male gametogenesis very early in the process- after the initial rounding up stage but before DNA replication and appearance of organised microtubule structures. Molecules in Cluster C appeared to arrest male gametogenesis subsequent to Cluster D. Two phenotypes were apparent in this cluster–one phenotype resembled Cluster D, with round cells and diffuse tubulin staining, however often the cells possessed elongated or more than one focal point of DNA within the cell suggesting that DNA replication had partially occurred. The other phenotype present in Cluster C showed defined microtubule structures surrounding concentrated areas of DNA staining, suggesting that DNA replication had perhaps progressed further and that gamete flagellae were forming prior to arrest. Cluster E appears to represent those molecules that arrested male gametogenesis very late in the process. All members induced phenotypes with large areas of DNA staining indicative of completed replication and clearly defined microtubule structures reminiscent of mature male gametes close to egress. These molecules likely inhibit the final maturation of the male gametes or prevent exit from the surrounding erythrocyte (which is not visualised in this study). Cluster B was heterogeneous, possessing molecules that appeared to arrest male gametogenesis at differing timepoints, as evidenced by molecules maintaining the falciform gametocyte shape and some inducing rounding up. The quantity of DNA staining and distribution within the cells was also not consistent, however all molecules in Cluster B induced clear microtubule structures. These molecules may act upon processes responsible for "normal" microtubule organisation/cell shape whilst having variable effects on DNA replication.

Molecules with similar chemical structures have a greater probability of sharing the same mode of action, therefore displaying the same phenotype and occupying the same phenotypic cluster. In validation of this hypothesis, the trisubstituted imidazole compounds TC41, TC27, TC04 and TC13 all gave an unactivated phenotype consistent with total arrest of gametogenesis and were part of Cluster A. Furthermore, N-4HCS scaffold molecules DDD01035881 and DDD01035879 arrested male gametogenesis late and were both members of Cluster E (**Fig 5**). One striking phenotype that emerged was shared by TC15 and TC18 (**Fig 5**) which have similar chemical structures (**Fig 2** and **S3 Table**). Unusually, both appear to result in male gametocytes that retain their falciform shape but have a condensed mass of unreplicated DNA and several thin microtubule structures.

Interestingly, Clusters A and D (**Fig 5**) that represented those molecules that either arrested gametogenesis before induction or those we hypothesise to arrest very early in the process

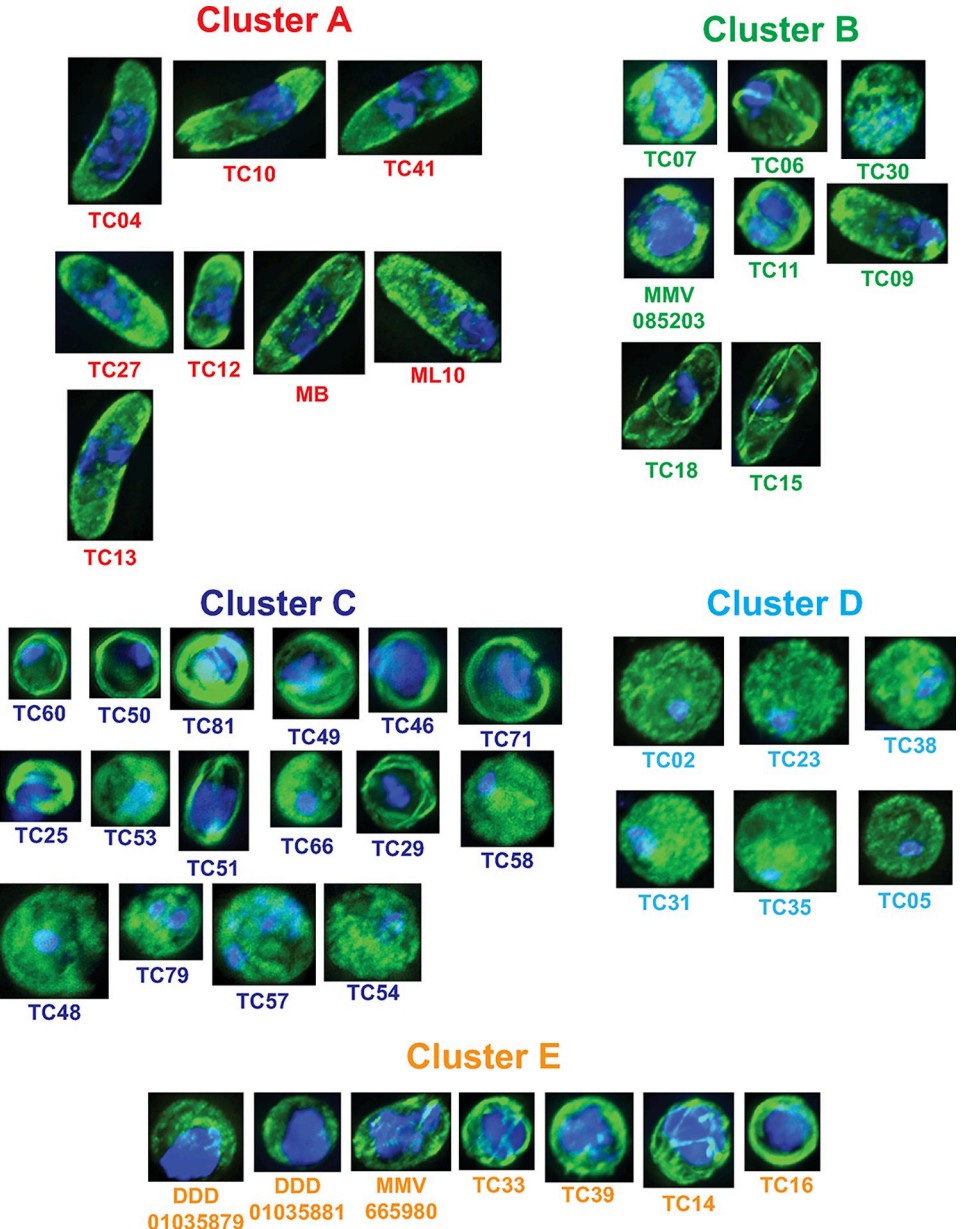

**Fig 5. Assignment of drugs to phenotypic clusters based upon principle component analysis and k-means clustering.** The cluster assignment of cells treated with each drug from Fig 4 was analysed by principle component analysis (PCA). Clusters were assigned by k-means clustering and 5 clusters (A-E) was found to be optimal using the Elbow method. Images indicate exemplar cellular phenotypes for each drug treatment. Scale bar = 3μm.

were found to all be active against both male and female gametocytes in the original screen (**Fig 2**). This is suggests that biological processes active before or early in gametogenesis are similar in both sexes. In contrast, molecules in Cluster C (**Fig 5**) that arrested male gametogenesis mid or late in the process showed a mix of male specific and dual activity and six out of seven "late" acting molecules in Cluster E were male-specific. This suggests that as gametogenesis progresses, signalling and metabolic pathways in male and female gametocytes increasingly diverge with time since induction.

## 5. Comparing the activity profile of selected molecules using phenotypic imaging

Contraceptive molecules block transmission by acting on cellular targets that are only directly active (or essential), during the process of male gametogenesis [19]. Therefore, it could be expected that the phenotype they elicit would change or disappear if gametocytes are exposed to the molecules progressively later after male gametogenesis has been induced. Additionally, molecules with promiscuous activity may show different inhibition phenotypes depending on timing of exposure during gametogenesis, linked to when their targets have critical functions.

To gain greater understanding of their activity, selected molecules displaying different inhibition phenotypes in the initial screen were re-tested. Rather than subjecting gametocytes to a 24h preincubation prior to inducing gametogenesis, compounds were added at varying timepoints after triggering male gametogenesis. Samples were then fixed 20 min post-induction, imaged and analysed similarly using the PhIDDLI pipeline.

When preincubated with gametocytes 24h prior to induction in the primary screen, TC11 (Cluster B), TC14 (Cluster E), and TC16 (Cluster E) caused arrest late in male gametogenesis (**Fig 5**). Similarly, exflagellation was substantially reduced if all three compounds were added concurrent with triggering male gametogenesis or 2 min later (**Fig 6C**). In contrast, if added after this, TC14 and TC16 were not inhibitory and exflagellation proceeded normally when compared to a parallel DMSO-treated sample. However, TC11 inhibited exflagellation even if added 10 min after induction.

Examining other phenotypic clusters identified by the PhIDDLI pipeline, we identified clusters containing cells with small faint DAPI staining and clusters with large strong DAPI staining (**Fig 6A and 6B**)–indicative of cells that had not undergone DNA replication and those that had, respectively. If added during the first 2 min of male gametogenesis, TC11 and TC16 treatment gave markedly reduced amounts of cells showing evidence of DNA replication. For example, if added concurrent with induction, only 18.6% and 21.7% of cells (TC11 and TC16 respectively) showed apparent DNA replication, compared to the DMSO control with 61.4% of cells assigned to the DNA replicated clusters (**Fig 6C**).

If added after this, corresponding with when DNA replication would be expected to already be complete during "normal" male gametogenesis, the number of TC11 and TC16-treated cells showing DNA replication was similar to the DMSO control. Surprisingly, treatments with TC14 had minimal impact on the proportion of cells with replicated DNA at all timepoints.

Taken together, this suggests that all three compounds act through distinct mechanisms of action to arrest male gametogenesis late in development. TC11 appears to elicit two phenotypic responses. If added early, cells arrest before DNA replication, whereas if added late, cells arrest after DNA replication and before exflagellation. This suggests either TC11 targets a pathway essential to the continued progression of male gametogenesis outside of the DNA replication pathway, or that the molecule shows polypharmacology and has two molecular targets.

A previous study showed that TC11 is a kinase inhibitor and was active against recombinant cyclin-dependent protein kinase 6 (PfPK6) and calcium-dependent protein kinase 1 (PfCDPK1) [22]. In male gametocytes, genetic deletion of PfCDPK1 arrests gametogenesis very late, preventing exflagellation but still enabling DNA replication and formation of internalised flagellae [28]. This would match the "late" inhibition component of the TC11 phenotype. PfPK6 currently has no described function in gametocytes but transcriptomic studies have showed PfPK6 transcripts are enriched in male gametocytes [28] and similar cyclin-dependent kinases are known to regulate the cell cycle in other organisms. In light of this, it is tempting to speculate that the "early" phenotype of TC11 could be due to inhibition of PfPK6 activity, which may regulate DNA replication. In contrast, TC14 appears to target a process

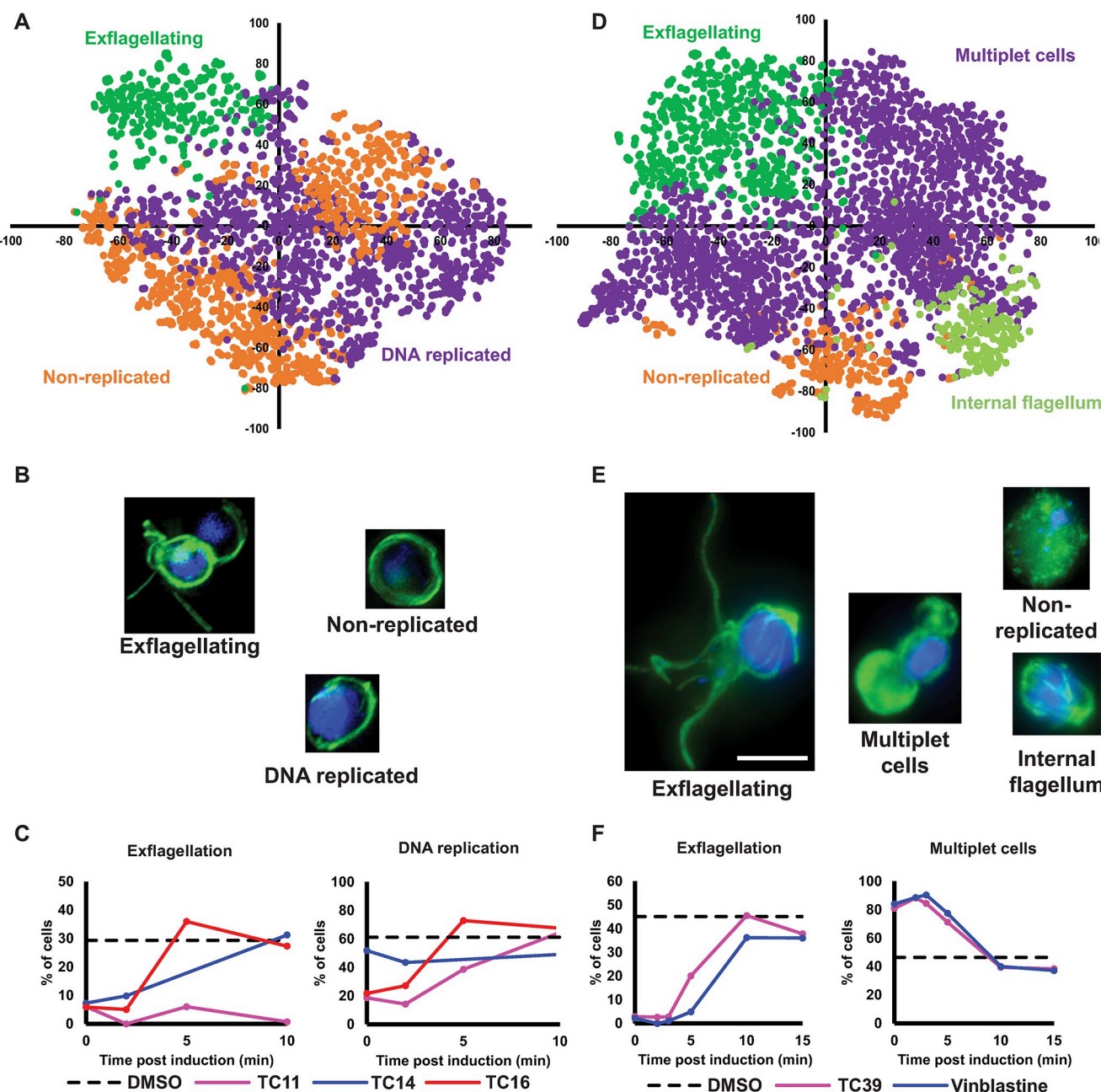

**Fig 6. Comparing how male gametogenesis changes phenotypically when cells are treated with compounds at different times post-induction.** (A-C) Cells were treated at 0, 2, 5 or 10 min post-induction of gametogenesis with early-acting TC11, TC14 and TC16. Resultant phenotypes were then assessed at 20 min post-induction when a parallel DMSO treated control showed maximal levels of exflagellation. (D-F) The phenotype of late-acting TC39 and microtubule depolymeriser vinblastine were evaluated if cells were similarly treated 0, 2, 3, 5 or 10 min post-induction of gametogenesis. (A + D) PhIDDLI plots of clusters of identified cellular phenotypes that were manually assigned after visual inspection. (B + E) Representative cells from each identified cluster. (C + F) Quantification of how cells from each compound treatment change phenotypic cluster depending on timing of compound treatment. Scale bar = 4μm.

that is critical only in the early stages of male gametogenesis and that is found in a pathway independent of DNA replication. Displaying further phenotypic diversity, similar to TC11, TC16 reduced the number of cells undergoing DNA replication if added in the first 5 min of gametogenesis; however, unlike TC11, it was not active thereafter.

TC39 (also known as TCMDC-123880) has recently been reported to affect cytoplasmic microtubule organisation in the rodent malaria species *Plasmodium berghei* male gametocytes during gametogenesis [10] and so its activity over male gametogenesis was also studied in comparison with a known inhibitor of microtubule assembly vinblastine (**Fig 6D-6F**). Both molecules strongly inhibited exflagellation if added between 0 and 5 min post induction of male gametogenesis. Interestingly, correlating with this decrease in exflagellation was the appearance of phenotypic clusters containing what appear to be multi-lobed "multiplet" cells, with either one lobe or both containing DNA staining. This phenotype is very striking and reminiscent of a recently reported phenotype induced by the putatively Pfs16-targeting N-4HCS scaffold which is observed when male gametocytes are treated with compounds, such as DDD01035881, within the first 4–5 min of male gametogenesis [9]. Indeed, in *P. berghei*, the Pfs16 orthologue has been linked to maintenance of correct microtubule organisation in the developing male gamete flagellum [29], thus a linkage between the phenotypes of molecules targeting Pfs16 and microtubule organisation is plausible.

In the primary screen with 24 h preincubation, treatment with TC02, TC05, and TC23 all gave phenotypes typified by round cells with diffuse tubulin staining and a small, concentrated nucleus suggestive of inhibited DNA replication and arrest early in gametogenesis (**Fig 5; cluster D**). In support of this, treating with TC02 or TC05 concurrent with triggering gametogenesis (or at any other point) had little-to-no impact on exflagellation (**Fig 7C**). In contrast, TC23 partially inhibited exflagellation if added concurrent with triggering gametogenesis (19.8% of cells exflagellating compared to 43.7% in the DMSO control), but not at any other timepoint tested. Taken together, the lack of activity profile of these early acting molecules suggests that these molecules require a period of incubation prior to induction of gametogenesis to accumulate in sufficient quantities in the gametocyte to elicit their effect. Alternatively they could also act upon targets upstream in the gametocyte that only manifest with a phenotype once gametogenesis is induced.

The trisubstituted imidazoles TC04, TC13, and TC41 that completely arrested the induction of male gametogenesis in the initial screen (**Fig 5, cluster A**) were evaluated in the time-course assay in parallel with ML10 –a known PfPKG inhibitor. If added concurrent with triggering gametogenesis, all molecules gave a near total arrest of gametogenesis (as reflected by lack of cells displaying a "late" phenotype") (**Fig 7F**) and most showed an unactivated phenotype. All molecules showed progressively reduced activity at later timepoints, with most cells displaying a "late" phenotype if compound was added 3–5 min after induction of gametogenesis. Interestingly, fully formed exflagellation centres were virtually absent from samples treated at 10 min with TC04 (only 0.4% of cells), TC13 (2.7%), and TC41 (1.1%), with the "late" phenotype predominating (86.3%, 83.3%, and 84.9%, respectively) (**Fig 7F**). This contrasts with ML10, which whilst showing reduced exflagellation compared to the control DMSO-treated sample (5.9% compared to 17.2%, respectively), still showed notable exflagellation (**Fig 7F**). Taken together, this suggests that whilst ML10, TC04, TC13, and TC41 can block the initial activation of male gametogenesis, TC04, TC13, TC41, and (to a much lesser extent) ML10 have additional activity which prevents the final maturation of male gametocytes into exflagellating male gametes. Another trisubstituted imidazole, TCMDC-140369/ MMV030084, has been demonstrated to be PfPKG inhibitor, but has additional activity against Pf CDPK1 [21]. TC04, TC13, and TC41 all have reported activity against recombinant PfCDPK1 in an enzymatic assay, but only TC13 had activity against recombinant PfPKG [22]. Taken together, our data suggests that TC04, TC13, and TC41 likely target both PfPKG and PfCDPK1. If gametocytes are treated prior to induction of gametogenesis or soon after, the trisubstituted imidazoles inhibit PfPKG which ablates gametogenesis and therefore masks the cellular phenotype caused by inhibition of PfCDPK1. If gametocytes are treated after PfPKG

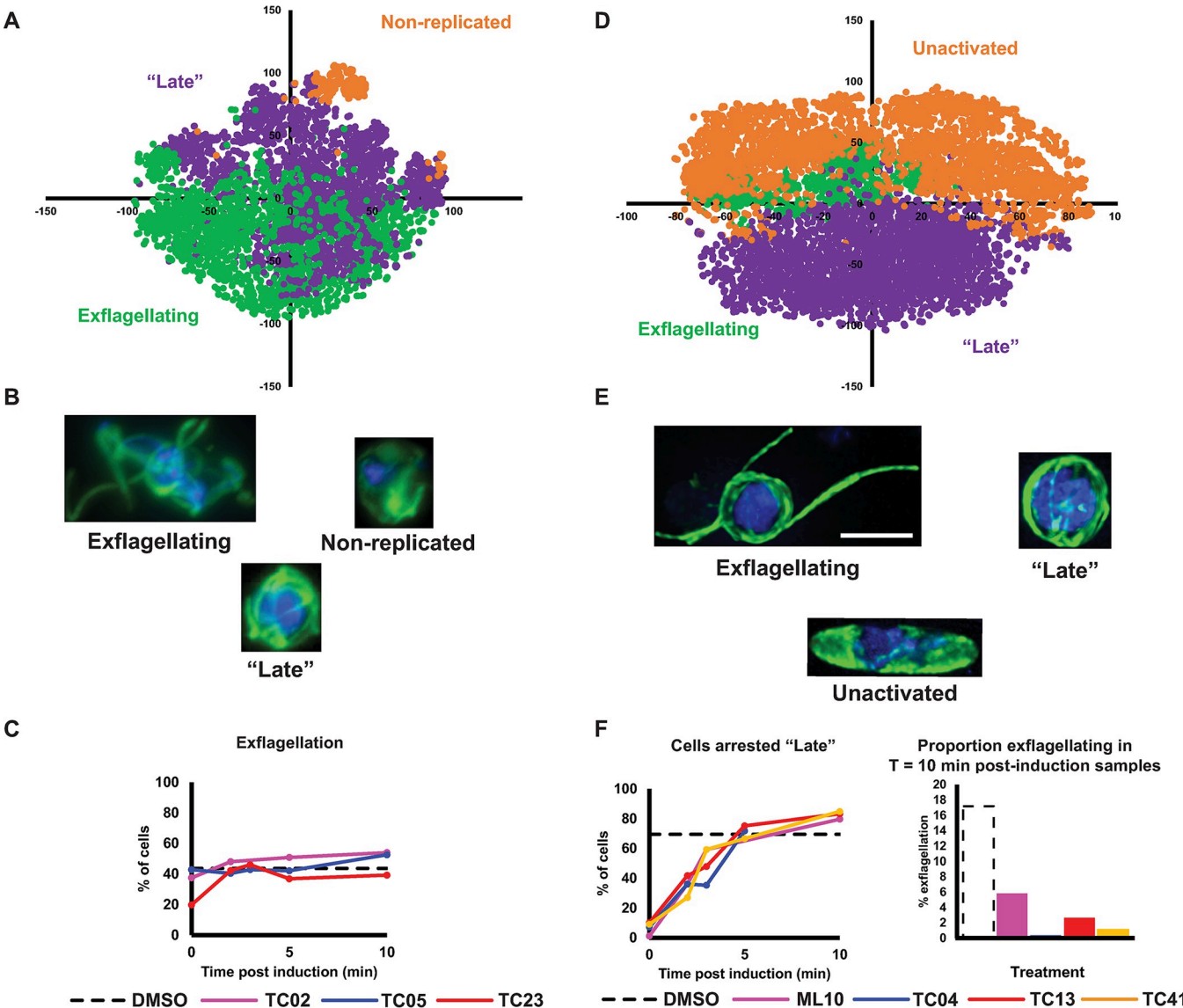

**Fig 7. Compounds targeting early male gametogenesis rapidly lose activity if administered later and compounds preventing induction of gametogenesis with a similar core structure show polypharmacology.** Cells were treated at 0, 2, 3, 5 or 10 min post-induction of gametogenesis and their resultant phenotypes studied at 20 min post-induction when a DMSO-treated control showed maximal levels of exflagellation. (A-C) TC02, TC05 and TC23 all showed early activity in the initial screen and so were compared together in the timecourse assay. (D-F) TC04, TC13 and TC41 which completely prevented induction of gametogenesis in the initial screen were evaluated with a known PKG inhibitor, ML-10. (A + D) PhIDDLI plots of clusters of identified cellular phenotypes that were manually assigned after visual inspection. (B + E) Representative cells from each identified cluster. (C + F) Quantification of how cells from each compound treatment change phenotypic cluster depending on timing of compound treatment. Scale bar = 4μm.

signalling has induced gametogenesis, the trisubstituted imidazoles display a phenotype reflecting their additional activity against PfCDPK1.

## Discussion

Mature stage V gametocytes reach the end of their development within the human host and await uptake from peripheral blood into the mosquito to continue their life cycle. Given that they cannot control when this event will take place, to maximise the chance of transmission mature gametocytes are adapted to live an extended life in a quiescent state. This results in

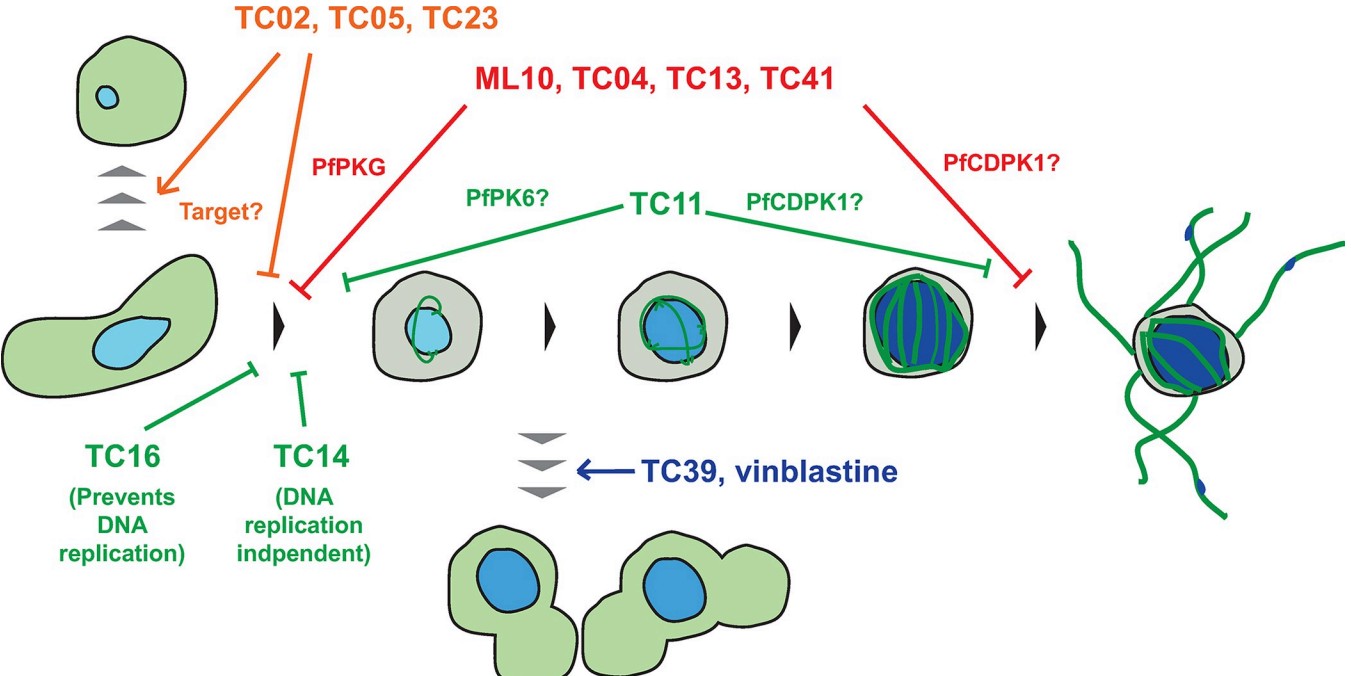

**Fig 8. A model summarising the observed activity of studied compounds in the timecourse assays.** Normal male gametogenesis involves the cell rounding up, replicating its genome three times, assembling up to eight microtubule-rich flagellae, and emergence of male gametes. Integrating data from this study, transmission-blocking molecules were either observed to halt male gametogenesis or generate cells with an entirely different phenotype.

their insensitivity to many antimalarial drugs that were designed against the fast-growing and proliferative asexual stages, targeting pathways that are absent, or not essential to the gametocyte.

Using our untargeted machine learning-aided phenotyping approach, we have taken the first steps towards systematically characterising the breadth of drug-targetable biology in the male gametocyte, identifying molecules interrupting development before, early or late in male gametogenesis (**Fig 8**). Tubulin arrangement, DNA content and overall cell shape are highly informative markers for male gametogenesis; however, visualising other essential processes, such as egress from the erythrocyte and secretion of osmophilic bodies, would also be equally enlightening. Since the PhIDDLI pipeline is agnostic to the images that it receives, future work could include phenotypic assays with different markers and/or transgenic knock down parasite lines to further enrich the dataset. In this context, a growing library of annotated phenotypes could be constructed as a reference for new transmission-blocking molecules and transgenic parasites.

Whilst not definitively identifying a specific mode of action, phenotypic imaging is a convenient tool to enable the researcher to rapidly generate evidence-based hypotheses for modes of action and aid the design of assays to test these efficiently. Furthermore, our timecourse assay format also permits finer detail in understanding the timing of action (or multiple sites of inhibition) of test molecules during male gametogenesis, and their use as tool compounds to dissect the signalling pathways controlling male gametogenesis. In this study, we highlight putatively promiscuous kinase inhibitors that likely target two (or more) *Plasmodium* kinases essential for male gametogenesis.

In all cases, these kinases are also essential for asexual development, leading to the potential for developing therapeutics that cure malaria, block transmission, and exhibit

polypharmacology. Molecules that have multiple targets are attractive as antimalarials as polypharmacology increases the barrier to resistance generation through the need to have direct mutation of two target genes. The challenge in developing such compounds further will be to increase their potency and selectivity for *P. falciparum* kinases over human kinases whilst still maintaining their polypharmacology.

## Materials and methods

### Compound management

Selected molecules from the TCAMS Library (comprising ~14,000 compounds previously reported to have activity against *P. falciparum* asexual parasites) were dissolved in 100% DMSO and dispensed directly onto screening-ready 384-well plates to give a final assay concentration of 2 μM and stored at -20˚C until needed. Active molecules selected for further study were supplied by GlaxoSmithKline as 10 mM DMSO stocks and stored at -20˚C. ML10 was kindly provided by Professor David Baker and methylene blue was kindly provided by the Medicines for Malaria Venture. All other control molecules were purchased from commercial vendors.

### Asexual and sexual parasite culture

*P. falciparum* NF54 (originally isolated from an imported malaria case in the Netherlands in the 1980s; BEI Resources, cat. No. MRA-1000) strain asexual parasites were maintained between 0.3–5% parasitaemia by regular passage in complete culture medium (RPMI supplemented with 25 mM HEPES, 50 μg ml$^{-1}$ hypoxanthine, 4.8 g l$^{-1}$ NaHCO$_3$, 2mM L-glutamine, 5% pooled type AB serum, 0.5% Albumax II (Gibco)) at 4% haematocrit in washed whole blood (National Blood and Transfusion Service or Centro de Transfusiones de la Comunidad de Madrid CTCM) under a 1% O$_2$/3% CO$_2$/96% N$_2$ environment. Gametocytes were induced by seeding culture flasks at 2% parasitaemia/4% haematocrit and subjecting them to daily medium changes for 14 days whilst maintaining a temperature of 37˚C at all times [17]. On Day 14, the maturity of gametocytes was assessed by quantifying male gametogenesis (exflagellation). A 20 μl sample of gametocyte culture was mixed with equal volume of ookinete medium (complete culture medium supplemented with 0.1 μM xanthurenic acid) at room temperature and transferred to a haemocytometer. After 20 min at room temperature, exflagellation was accurately quantified as described in Delves et al. 2016 [17]. Cultures possessing >0.2% exflagellation (including uninfected erythrocytes) were selected for downstream assays.

### *P. falciparum* Dual Gamete Formation Assay (Pf DGFA)

The Pf DGFA was performed as described in Delves et al. 2016 [17]. Briefly, gametocyte cultures were diluted to 12.5 million cells ml$^{-1}$ and 50 μl was rapidly dispensed into every well of a 384-well plate containing test compounds which had been prewarmed to 37˚C. Plates were then incubated for 24 h at 37˚C in a humidified box under a 1% O$_2$/3% CO$_2$/96% N$_2$ gas mix. Gametogenesis was triggered by addition of 10 μl per well of ookinete medium + 0.27 μg ml$^{-1}$ Cy3-labelled anti-Pfs25 antibody. Plates were incubated on a metal block at 4˚C for 4 min to ensure even cooling, then 28˚C for a further 4 min. Exflagellation was recorded by capturing 10 frames over 1 s under brightfield illumination using a x4 objective with a 1.5x zoom (effectively x6) attached to an automated Nikon Ti fluorescence microscope. After recording exflagellation, plates were incubated for a further 24 h in the dark at 28˚C to ensure maximal expression of Pfs25 by female gametes. The plates were then returned to the microscope and female gametes imaged at x6 using fluorescence microscopy.

Exflagellation centres and female gametes were quantified using custom image analysis pipelines developed within ICY Bioimage Analysis (https://icy.bioimageanalysis.org/). Percent inhibition of male/female gametogenesis were calculated with reference to negative (DMSO) control and positive (20 μM gentian violet) wells.

## Cheminformatics

Pf DGFA activity was compared to reported activity in other published transmission-blocking screens [6,7,20]. Potential compound targets were compiled from previously reported activity against asexual parasites and select parasite kinases [16,22]. Compounds were clustered by chemical similarity in DataWarrior [30] using the FragFp algorithm and visualised with Cytoscape 3.7.0 (https://cytoscape.org/).

## PhiDDLI screening assay

Mature gametocyte cultures were divided into 100 μl "microcultures" in pre-warmed 1.5 ml tubes also containing 100 μl complete culture medium with test compounds at a final concentration of $10xIC_{50}$ or 0.5% DMSO as a solvent control. Tubes were gassed and incubated at 37˚C for 24 h. The entire compound library was screened once, spread over four different experiments, each with two DMSO control tubes. An 8% formaldehyde solution was prepared by diluting 16% methanol-free formaldehyde in an equal volume of phosphate-buffered saline (PBS). In each experiment, one DMSO control tube was fixed with equal volume of 8% formaldehyde (4% final concentration) at 37˚C to provide an "unactivated" gametocyte control. Gametogenesis was then triggered in the remaining DMSO control tube and all compound tubes by adding 10 μl ookinete medium, and they were immediately incubated at room temperature. After 20 min (when exflagellation is maximal), all remaining tubes were fixed for 30 min.

Alternatively, if compound action over time was to be assessed, 1.5ml tubes with appropriate dilutions of test compounds were prepared for each timepoint. Gametogenesis was then triggered in the mature gametocyte culture with ookinete medium and temperature decrease from 37˚C. At appropriate times, 200 μl of triggered culture was sampled and mixed with test compound. All samples were then allowed to continue gametogenesis and fixed as described previously with paraformaldehyde after 20 min (when a DMSO-treated sample could be expected to be exflagellating).

100 μl of the fixed cell suspensions were added to wells of a 24 well plate containing 1ml PBS and a poly-l-lysine-coated glass coverslip. Cells were allowed to settle and adhere to the coverslip overnight at 4˚C before coverslips were transferred to a hydrophobic membrane (Parafilm) for immunofluorescence staining in a humidified, dark box. Cells were permeabilised with 0.1% Triton-X100 in PBS for 10 min before washing in PBS three times. Non-specific binding was blocked by incubating coverslips with PBS + 10% fetal bovine serum (FBS) (blocking buffer) for 1 h. Gametocytes and microgametes were visualised by labelling with an anti-alpha tubulin antibody (raised from chicken embryo brain tubulin but cross-reacts with many other species) (clone DM1A (Sigma)) 1:500 dilution in blocking buffer for 1 h, washed as before, and then stained with an anti-mouse Alexa488 secondary antibody (Thermo-Fisher) 1:1,000 dilution in blocking buffer for a further 1 h. After a final series of washes, the coverslips were mounted on glass slides with VectaShield mountant (Vector Laboratories) containing 4′,6-diamidino-2-phenylindole (DAPI) to visualise cellular DNA.

## Fluorescence microscopy and image processing

All slides were imaged using a Nikon Ti fluorescence microscope using an x100 oil objective and 1.5x zoom. For each sample, tubulin-postive, DAPI-positive stained gametocytes/gametes

were imaged at random. Z-stack images with 0.3μm slices were captured for each field of view and images deconvolved and transformed by maximum intensity projection within the Nikon NIS Elements software. Within raw images, cells were initially identified by an automated analysis pipeline developed in ICY Bioimage Analysis based upon the intensity of the tubulin staining captured which assigns a region of interest (ROI) around each cell. All ROIs were reviewed by a trained operator and adjusted if necessary to reflect the true outline of the cell (for example if two cells touching had been allocated the same ROI). Raw images contained between 1 and 8 extractable cells.

## PhIDDLI pipeline

Code for the PhIDDLI pipeline was written in Python 3.8 and executed by command line within a 64-bit Ubuntu virtual machine running within Oracle VM VirtualBox 6.1. Data was visualised as an interactive web page displayed in Mozilla Firefox browser and also exported as a comma separated values (csv) file for further analysis. Principle component analysis of compound phenotypes (**Fig 5**) was performed using ClustViz (https://biit.cs.ut.ee/clustvis/) using ln(x+1) transformation, and k-means clustering of the resultant PCA data was coded in R and executed in R Studio version 1.1.456 to identify optimum cluster using the elbow method.

## Supporting information

**S1 Table. Screening of molecules in the PfDGFA at 2μM with 24h preincubation with gametocytes prior to induction of gametogenesis.** Hits were selected based upon >50% inhibition in either the male or female readout in at least 3 independent biological replicates. (Activity: D (MF) = dual; M = male-biased; F = female-biased)
(XLSX)

**S2 Table. The PfDGFA potency of active compounds were determined and compared to cytotoxic activity against HepG2 cells (Activity: D = dual; M = male-biased; F = female-biased)**
(XLSX)

**S3 Table. Summary of the molecules characterised in this study.** Data presented here includes the SMILES codes of each molecule, concentration tested in the PhIDDLI screen, target prediction based upon literature search, activity in previously reported assays and reported cytotoxic activity against HepG2 cells.
(XLSX)

**S1 Fig. Selected cells from Cluster 6 (Fig 4).** Cluster 6 represents the convergence of many clusters and consequently there was significant diversity of cell morphology within the cluster. The majority were irregular-shaped but showed ordered microtuble organisation. Bar = 3μm.
(PDF)

**S2 Fig. The distribution of cells from individual drug treatments visualised within the entire dataset.**
(PDF)

**S3 Fig. Principle component analysis (PCA) plot of the first two principle components (representing 56.8% of the total variance) of the cluster assignments from Fig 4.** The percentage of cells from each drug treatment falling into each of the 9 identified clusters was compared by PCA. All nine computed principle components were then used to cluster each drug phenotype by k-means clustering and the elbow method which determined 5 clusters was

optimal. Cluster assignment is summarised in Fig 5. Bar = 3μm.
(PDF)

## Author Contributions

**Conceptualization:** Maurizio Gioli, Marco Santelli, Edgar Whitley, Michael J. Delves.

**Data curation:** Michael J. Delves.

**Formal analysis:** Francisco-Javier Gamo, Michael J. Delves.

**Funding acquisition:** Maurizio Gioli, Marco Santelli, Edgar Whitley, Michael J. Delves.

**Investigation:** Celia Miguel-Blanco, Penny C. Sparkes, Michael J. Delves.

**Methodology:** Oleksiy Tsebriy, Andrii Khomiak, Maurizio Gioli, Michael J. Delves.

**Project administration:** Michael J. Delves.

**Resources:** Penny C. Sparkes, Francisco-Javier Gamo.

**Software:** Oleksiy Tsebriy, Andrii Khomiak, Marco Santelli.

**Supervision:** Marco Santelli, Edgar Whitley, Michael J. Delves.

**Validation:** Francisco-Javier Gamo, Michael J. Delves.

**Visualization:** Oleksiy Tsebriy, Michael J. Delves.

**Writing – original draft:** Michael J. Delves.

**Writing – review & editing:** Celia Miguel-Blanco, Maurizio Gioli, Edgar Whitley, Francisco-Javier Gamo, Michael J. Delves.

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
