## [Decision Letter · Decision Letter 0]

26 Jul 2023

Dear Dr Delves,

Thank you very much for submitting your manuscript "Machine learning-based phenotypic imaging to characterise the targetable biology of Plasmodium falciparum male gametocytes for the development of transmission-blocking antimalarials" for consideration at PLOS Pathogens. As with all papers reviewed by the journal, your manuscript was reviewed by members of the editorial board and by several independent reviewers. In light of the reviews (below this email), we would like to invite the resubmission of a significantly-revised version that takes into account the reviewers' comments. The reviewers see value in this work but their comments indicate that the manuscript would benefit from some significant additional analyses to more firmly establish the robustness of the pipeline.

We cannot make any decision about publication until we have seen the revised manuscript and your response to the reviewers' comments. Your revised manuscript is also likely to be sent to reviewers for further evaluation.

Sincerely,

Mostafa Zamanian

Guest Editor

PLOS Pathogens

James Collins III

Section Editor

PLOS Pathogens

Kasturi Haldar

Editor-in-Chief

PLOS Pathogens

orcid.org/0000-0001-5065-158X

Michael Malim

Editor-in-Chief

PLOS Pathogens

orcid.org/0000-0002-7699-2064

Reviewer's Responses to Questions

**Part I - Summary**

Reviewer #1: The manuscript by Tsebriy et al. outlines and applies a custom modular pipeline based on open-source software to segment and characterise sexual stage Plasmodium falciparum parasites. Using known and uncharacterised compounds they demonstrate the workflow clusters these compounds together – based on their modes of action. They go on to show how parasite activation – particularly male gametocyte activation - is temporally sensitive to some of these drug candidates. This demonstrates the workflow can be used to define effects of compounds of gametocytogenesis through time.

This work demonstrates a timely piece of research which would be of great value to the scientific community. However, the manuscript would benefit from clarification in certain areas which are outlined next.

Reviewer #2: The manuscript by Tsebriy and colleagues proposes the use of a machine learned (deep neural network) based pipeline (PhIDDLI) to classify images of P. falciparum male gametocytes/gametes treated with antimalarial compounds as a way to associate specific morphological features of treated parasites to chemical scaffolds to help elucidating compound(s) mechanisms of action.

The work is inspired by the approach described in reference 15, in which images capturing the continuum of cellular modifications occurring during the development of asexual parasites were subjected to a similar deep neural network based classification and the algorithm was then tested on parasites treated with compounds active on specific cellular targets.

I have some fundamental concerns on the pipeline presented here, starting from some baseline methodology aspects (Part II of the review).

Even though morphology alterations caused by drug treatment are, as authors state, just a preliminary glimpse into modes of actions, the authors’ effort to develop artificial intelligence approaches for the desirable scaling up of transmission blocking drug discovery is indeed important. However the claim that PhIDDLI is adequate to the task and that can already contribute to this goal on other systems is an overstatement. Analysis of some of the results of the pipeline presented here casts doubts on machine learning performance and suggests that visual inspection still plays an important role in this analysis. For instance, the parasite morphologies shown in Figures 6 and 7, respectively named “multiplets” and “coffee beans” do not seem to be at all represented in the PhIDDLI features of Figures 3 and 4.

**Part II – Major Issues: Key Experiments Required for Acceptance**

Reviewer #1: 1. As the authors use maximum intensity projection, 3D information is lost. The authors should comment on whether this reduces sensitivity of the classification. For example, in Figure 4, the example images of Cluster 11 and Cluster 7 or Cluster 4 and Cluster 9 could easily fall into the other category. Can the authors show images belonging to specific clusters that have been manually altered (scaled, flipped, rotated) are still classified in those same clusters.

2. As the EfficientNet-B1 model is pre-trained for image processing, it may be missing or over-fitting certain data, can the authors comment on this. Additionally, EfficientNet-B1 achieves under 78% accuracy, while B7 now achieves 84% accuracy – can the authors run analysis non the most current model for greater robustness?

Reviewer #2: The first methodology concern is related to the nature and the number of the fluorescent markers used to stain cellular structures. Use of only two probes, i.e. DAPI for parasite nuclear staining and an antibody against tubulin (on which further comments are below), fails to reveal additional important features which could be harnessed for classification such as mitochondrial and erythrocyte surface staining. In fact, visual inspection of the features obtained before and after dimensionality reduction (Figure 3) easily spots virtually indistinguishable morphological patterns.

The second concern is that no mention is present in the manuscript of how PhIDDLI deals with the fact that the majority of parasites imaged in the control and treated wells are female gametocytes/gametes. In the NF54 parasite line used here the gametocyte sex ratio (proportion of males) is strongly female biased, around 1:8 1:10. In this respect, whilst the rationale of labelling microtubules appears logical, as indeed these structures are assembled only in male gametogenesis, the anti-alpha tubulin antibody used here is not gametocyte sex specific. In fact, the statement that the antibody is against alpha-tubulin II (section 2, Results) is incorrect, as the reagent is a monoclonal antibody labelling tubulin across several mammalian species (Materials and Methods), and ambiguous, as it is reminiscent of the P. falciparum protein tubulin alpha II. Antibodies against the latter, nevertheless, are not male gametocyte specific, but show slightly different staining patterns in the cytoplasm of female and male late gametocytes (https://pubmed.ncbi.nlm.nih.gov/21209927/).

The third concern is that I would have expected that a baseline step of the machine learning approach was dedicated to classify the many steps of a normal, unperturbed male gametogenesis, that authors rightly describe to occur in vitro as “an inefficient process that not every male gametocyte completes after stimulation”. Introduction of a training set of images encompassing the events of a normal, yet inefficient male gametogenesis, would likely help to specifically identify and classify alteration in parasite morphology induced by compounds.

**Part III – Minor Issues: Editorial and Data Presentation Modifications**

Reviewer #1: Additional issues, that are less impactful to the integrity of the work but should nonetheless be addressed include:

3. Page 6: Clarity is needed on how efficient the acquisition to analysis is. ‘Images (n = 1207) and bounding boxes… and extraction of individual cells (n = 2446)’. Does this mean only 2 cells were extracted from each captured image?

4. Scale bars should be added to Figures containing images.

5. Images were extracted for ‘Tubulin-bright cells’, what if the drug ablated microtubules completely? This data would be lost, can the authors comment on this?

6. Figure 3: Please make axis labels in any final versions readable.

7. Figure 6 title: remove ‘the’: ‘Comparing the how male…’

8. Figure 7: It seems the ‘Coffee bean’ phenotype (Pink cluster) is distributed in both the extreme positive and negative of the x axis. Why are they separated so widely?

9. Page 12: 1st paragraph, shorten the last sentence which is 5 lines in length.

10. Page 12: 2nd paragraph, Clarify which Figure they are referring to when citing ‘(Cluster A)’

11. Page 12: Final section before Discussion – Making the summary of compounds into a table would make this much easier to navigate.

Reviewer #2: With the above fundamental concerns, minor questions on details of the pipeline are

1- the reason for having 1000 features extracted by the staring set of 2446 cells from the 1027 images;

2- the parameters or rationale leading to set to 15 the number of clusters in the k-cluster analysis means;

3- what is the percent of variance associated to the PCA components plotted in Fig. 5;

4- what images were used to pre-train the EffcientNet-B1 module, the open source deep neural network of PhIDDLI.

PLOS authors have the option to publish the peer review history of their article (what does this mean?). If published, this will include your full peer review and any attached files.

Reviewer #1: No

Reviewer #2: No
---

## [Decision Letter · Decision Letter 1]

25 Sep 2023

Dear Dr Delves,

We are pleased to inform you that your manuscript 'Machine learning-based phenotypic imaging to characterise the targetable biology of Plasmodium falciparum male gametocytes for the development of transmission-blocking antimalarials' has been provisionally accepted for publication in PLOS Pathogens.

Best regards,

Mostafa Zamanian

Guest Editor

PLOS Pathogens

James Collins III

Section Editor

PLOS Pathogens

Kasturi Haldar

Editor-in-Chief

PLOS Pathogens

orcid.org/0000-0001-5065-158X

Michael Malim

Editor-in-Chief

PLOS Pathogens

orcid.org/0000-0002-7699-2064

Reviewer Comments (if any, and for reference):

Reviewer's Responses to Questions

**Part I - Summary**

Reviewer #1: The authors have provided the required changes or justifications to the manuscript based on Reviewer comments.

**Part II – Major Issues: Key Experiments Required for Acceptance**

Reviewer #1: N/A

**Part III – Minor Issues: Editorial and Data Presentation Modifications**

Reviewer #1: N/A

PLOS authors have the option to publish the peer review history of their article (what does this mean?). If published, this will include your full peer review and any attached files.

Reviewer #1: No

---

## [Editor Report · Acceptance letter]

3 Oct 2023

Dear Dr Delves,

We are delighted to inform you that your manuscript, "Machine learning-based phenotypic imaging to characterise the targetable biology of *Plasmodium falciparum* male gametocytes for the development of transmission-blocking antimalarials," has been formally accepted for publication in PLOS Pathogens.

Best regards,

Kasturi Haldar

Editor-in-Chief

PLOS Pathogens

orcid.org/0000-0001-5065-158X

Michael Malim

Editor-in-Chief

PLOS Pathogens

orcid.org/0000-0002-7699-2064